# A Global Study on the Correlates of Gross Domestic Product (GDP) and COVID-19 Vaccine Distribution

**DOI:** 10.3390/vaccines10020266

**Published:** 2022-02-10

**Authors:** Palash Basak, Tanvir Abir, Abdullah Al Mamun, Noor Raihani Zainol, Mansura Khanam, Md. Rashidul Haque, Abul Hasnat Milton, Kingsley Emwinyore Agho

**Affiliations:** 1School of Environment and Life Sciences (Environmental Science and Management), University of Newcastle, Callaghan, NSW 2308, Australia; Palash.Basak@newcastle.edu.au; 2Faculty of Business and Entrepreneurship, Daffodil International University, Daffodil Smart City, Dhaka 1341, Bangladesh; t.abir73@gmail.com; 3UKM—Graduate School of Business, Universiti Kebangsaan Malaysia, Bangi 43600, Malaysia; mamun7793@gmail.com; 4Faculty of Entrepreneurship and Business, University Malaysia Kelantan, Kota Bharu 16100, Malaysia; raihani@umk.edu.my; 5International Centre for sDiarrhoeal Disease Research, Bangladesh (Icddrb), GPO BOX 128, 68, Shaheed Tajuddin Ahmed Sarani, Dhaka 1212, Bangladesh; mansura@icddrb.org; 6Department of Psychiatry, Sir Salimullah Medical College, Dhaka 1206, Bangladesh; rana.dmc@gmail.com; 7Research International, Bangladesh & Research and Training International, Newcastle, NSW 2290, Australia; mhasnat@retin.com.au; 8School of Health Sciences, Campbelltown Campus, Western Sydney University, Penrith, NSW 2571, Australia; 9Translational Health Research Institute (THRI), School of Medicine, Campbelltown Campus, Western Sydney University, Penrith, NSW 2571, Australia; 10African Vision Research Institute (AVRI), University of KwaZulu-Natal, Durban 4041, South Africa

**Keywords:** COVID-19 vaccination, GDP, public health, high-income countries, developing countries

## Abstract

This study aimed to explore the association between the GDP of various countries and the progress of COVID-19 vaccinations; to explore how the global pattern holds in the continents, and investigate the spatial distribution pattern of COVID-19 vaccination progress for all countries. We have used consolidated data on COVID-19 vaccination and GDP from Our World in Data, an open-access data source. Data analysis and visualization were performed in R-Studio. There was a strong linear association between per capita income and the proportion of people vaccinated in countries with populations of one million or more. GDP per capita accounts for a 50% variation in the vaccination rate across the nations. Our assessments revealed that the global pattern holds in every continent. Rich European and North-American countries are most protected against COVID-19. Less developed African countries barely initiated a vaccination program. There is a significant disparity among Asian countries. The security of wealthier nations (vaccinated their citizens) cannot be guaranteed unless adequate vaccination covers the less affluent countries. Therefore, the global community should undertake initiatives to speed up the COVID-19 vaccination program in all countries of the world, irrespective of their wealth.

## 1. Introduction

The worldwide effort to create safe and effective COVID-19 vaccines has produced remarkable results, thanks in part to early, crucial investments in clinical discovery through initiatives like Operation Warp Speed [1]. These accomplishments demonstrate the benefits of consistent, extended funding for basic research and immunology; the scientific community was prepared to take action. Now, as the world is faced with a scarcity of vaccines, there is a depressing reality: As of 25 December 2021, 57.4% of the world population has received at least one dose of a COVID-19 vaccine [2]; approximately 8.95 billion doses have been administered globally, and 37.19 million are now administered each day [2]. However, only 8.3% of people from low-income countries have received at least one dose [2]. Vaccine distribution is still very low in many of the world’s poorest countries. Experts predicted that 20% of the population in low-resource areas would receive a vaccine in 2021 [3], but in reality, the actual achievement was much lower. The critical nature of investment in research notwithstanding, prolonged neglect of public health and global delivery strategies has rendered humankind unprepared to bring this pandemic to an end. Priority must be given to solving the complex bottlenecks in distributing and allocating newly approved vaccines [3]. As part of these efforts, vaccines must be produced in a safe, efficient, and timely manner. As a result of mistrust, misinformation, and historical legacies, vaccine adoption has been hindered [4]. Even wealthy countries have encountered formidable obstacles when implementing mass vaccination programs, and have made critical mistakes [3].

Aside from that, the early competitive procurement of vaccines by the United States and purchases by other high-income countries has led to the universal assumption that each country will be exclusively responsible for its population. When powerful countries secure vaccines and therapies at the expense of less-wealthy countries, it perpetuates a long history of shortsightedness, inefficiency, and death [3]. Developed or Industrialized countries are anxious to help with global vaccination, particularly for countries that require partnerships to warrant supply and delivery. Uncoordinated patches of immunity could also exacerbate the spread of escape variants [3]. As a result of these inequalities, vaccines and essential medications are treated as a market commodity instead of a public good in global health, and more widely in our global economy.

Similar policies have been implemented during pandemics of the past. Antiretroviral therapy was out of reach for most low-resource countries when HIV was at its zenith because of prohibitively high prices imposed by the pharmaceutical industry, as well as a belief among United Nations agencies and major donors that prevention should take precedence over treatment [3]. Inequities in access and health and economic well-being are exacerbated by the commodification of public goods in the global economy. As a moral and national security issue, eliminating critical constraints requires bold, decisive action to ensure the expansion of supply and delivery of COVID-19 vaccines. As part of the COVID-19 Vaccines Global Access (COVAX) program, which supplies vaccines to low- and middle-income countries, the United States, under the Biden administration, and the G7 nations have pledged support for global vaccine procurement. Still, this funding is insufficient [3]. Currently, COVAX plans to vaccinate at least 20 percent of participating countries’ populations by the year 2021. Even though this would be a significant achievement, it falls far short of the goal of quickly securing global herd immunity [5].

Even prior to any vaccine’s approval, high-income countries that constitute only a fraction of the global population had already placed orders for more than 50% of the projected early supply of doses for COVID-19 vaccines. By mid-August of 2020, the US had secured 800 million doses of at least six different vaccines in development; the UK had procured 340 million doses, with around 5 per capita; and the European Union (EU) and Japan had each placed orders for hundreds of millions of doses [6]. While the world’s wealthiest nations have booked enough doses of the best COVID-19 vaccines to immunise their individual populations numerous times [7], forecast global manufacturing capacity also implies limited and delayed access of low-income nations to this significant healthcare resource. This current study examined the association between national income and COVID-19 vaccination in countries with 1 million or more population. It highlights how vaccine nationalism in the context of COVID-19 has uncovered the risks of nationalist responses to a simultaneous global emergency and how it thus represents a pivotal moment in which the dynamics, courses, and directions of contemporary globalization processes must be crucially reflected on and acted upon.

## 2. Materials and Methods

### 2.1. Data

The Coronavirus Pandemic (COVID-19) dataset used in the research was downloaded from Our World in Data [2] at 7:43 p.m. on 25 December 2021, Australian Eastern Daylight Time (AEDT). Along with the vaccination rate, the dataset contained information about GDP. For exploring evidence of vaccine nationalism, both vaccination rate and GDP data were used. Countries with a population of less than one million were excluded from the study to focus on sizable nations. The dataset was also filtered for null values. A total of 153 countries with one million or more population were included in the study, and their data on GDP and vaccination were available.

### 2.2. Analysis

R-Studio Cloud (RStudio Workbench, Version 2021.09.1 Build 372.pro1 [8] was used for data gathering and reorganization and statistical analysis. Graphs were also constructed with the same application. Ref. [9] was used to prepare the map.

## 3. Results

Figure 1 shows a strong linear association between GDP per capita and the proportion of the people vaccinated (partially or fully) in 153 countries of the world (*r* = 0.79, *p* < 0.001), where data for both vaccination rates and GDP were available. GDP per capita (log_10_) explains about 63% variation in the vaccination rate across the countries. In general, the wealthier the nation is, the higher the vaccination rate. The top three countries with the highest vaccination rate are the United Arab Emirates (UAE), Portugal, and Chile. GDP per capita of these top-performing countries is more than US$22,000, and their vaccination rate is above 89%. In contrast, the bottom three countries are Burundi, the Democratic Republic of Congo (DRC), and Haiti. The GDP of these countries is less than US$1700, and their vaccination rate is 1% or less.

Similarly, a statistically significant correlation was also observed between GDP per capita (log_10_) and full vaccination rate, *r* = 0.82, *p* < 0.001, there is a linear association between these two variables (Figure 2). Countries like UAE, Portugal, and Singapore have already provided full vaccination to over 87% of people, while Burundi, DRC, and Chad barely immunized people with two doses (less than 0.5%). The data on partial and full vaccination rates are highly correlated, *r* = 0.98, *p* < 0.001 (Figure 3). The countries with higher partial vaccination also achieved higher full vaccination rates (detailed information of all countries are provided in Appendix A).

The global pattern of the association between GDP per capita and vaccination holds across continents (Figure 4 and Figure 5). The richer the country, the higher the vaccination rate is in every continent. The wealthy countries in Europe, North America, and Asia managed to protect a higher proportion of the population with the vaccine. As shown in Figure 4 and Figure 5, African countries have the lowest income. Except for a few, such as Mauritius and Morocco, all other countries in that continent are lagging in COVID-19 vaccination.

The highest level of disparity in GDP per capita and vaccination rate has been observed in Asia. While less-developed Asian countries like Yemen, Afghanistan, and Kyrgyzstan vaccinated less than 20% of their population, more than five countries in the continent achieved over 80% vaccination rates. India performed relatively better in immunizing people compared to its wealth, and the country has achieved a vaccination rate of approximately 60%. Being a country with modest GDP, Bangladesh follows the trend but has made better progress in recent months. About 53% of the population of Bangladesh has received at least one dose of COVID-19 vaccine so far. UAE performed better than Kuwait, even though their national income level is similar. Yemen performed the worst in of any country in Asia.

In Europe, countries with a lower GDP per capita, such as Moldova, Bulgaria, and Bosnia and Herzegovina, obtained 26% or lower vaccination rates. On the other hand, Portugal, Spain, and Denmark achieved over 80% vaccination rates, and their GDP per capita is relatively high. Twenty countries in Europe—including the United Kingdom, France, and Germany—attained 63% or greater vaccination rates, and their GDP is over US$24,000. In Oceania, less developed Papua New Guinea vaccinated about 3% of its people, while Australia and New Zealand vaccinated over 77% of theirs.

Canada (83%) and the United States (USA, 62%) performed the best among the North American countries. Countries like Haiti, Jamaica, and Guatemala managed to vaccinate less than 35% of their population in the same North America continent. In South America, Venezuela appears to have made less progress in vaccination in comparison to its wealth—it managed to vaccinate approximately 64% of its population. The rest of the countries in the continent achieved vaccination rates proportional to their GDP per capita, and Chile and Argentina in South America obtained over 82% vaccination rates.

The spatial distribution pattern reveals that African nations are least protected against COVID-19 (Figure 6). North American and European nations made the highest level of progress in vaccinating their population. Low-income African countries could not make much progress in the vaccination program during the last few months. There is a significant variation in the vaccination rate among Asian countries (Figure 5). However, the variable vaccination rate in Asia is explainable with the differences in income level.

## 4. Discussion

In this current study, we aimed to establish the association between the GDP of nations with COVID-19 vaccination rates. We found that, in general, the wealthier a country is, the higher the vaccination rate. In contrast, the bottom three countries are the Democratic Republic of Congo (DRC), Haiti, and Chad. The GDP of these countries is US$1700 or less, and their vaccination rate is lower than 1%. Two of the three countries, the DRC and Chad, are in Africa. It has thus been revealed with COVID-19 vaccination progress that humanity is divided into “haves” and “have-nots”. Developed countries secure their populations with vaccination, unlike their counterparts in the developing world, where vaccination rates are lower. The important takeaway message is that the COVID-19 vaccine rollout is uneven and that countries in the Global South are not performing as well as their Global North counterparts. Therefore, there are large country-level disparities in being able to achieve herd immunity, and efforts toward attaining global herd immunity are severely hampered. For instance, it has always been the desire of low- and middle-income countries to gain access to technological and medical advancement, including vaccinations and drugs. One unfortunate example of this scenario is the massive toll from HIV (Human Immunodeficiency Virus) and AIDS (Acquired Immunodeficiency Syndrome) from 1997 to 2007 in Africa, with an alarming record of 12 million deaths that devastated the continent [10]. While industrialized nations had made the drugs for the disease sufficiently accessible, there was still a surrender to this virus in African countries. An identical occurrence was noticed in the 2009 swine-flu epidemic [10]. Wealthy nations acquired the vaccine for the disease at excessive rates, whereas the economically less affluent nations were found wanting. In contrast to the higher-income nations which were able to implement lockdowns and institute physical distancing protocols, the LMICs have hardly been able to do the same, and consequently reared a blossoming population of more vulnerable people. This does not suggest that the richest countries are acting against the poorest ones. Furthermore, the authors do not suggest that the richest countries have to be involved in the poorest countries’ decisions, such as research investment, etc.

Furthermore, the infrastructure is wretched in these LMIC countries, and includes bad roads and housing, giving rise to a huge number of difficult-to-reach populations. In spite of this, and despite the fact that there is a limitation to their financial contributions to vaccine development, the world must include them in vaccination protocols towards a common enemy who has no respect for geography or social standing. Developing countries should be supported in their efforts toward “ensuring access to the COVID-19 vaccine by levelling the power dynamics that perpetuate inequality and fuel injustice” [11].

Although a large proportion of the population in many countries has been vaccinated against COVID-19, it is not the case across the world, where billions (mostly from the world’s poorest) are yet to receive their vaccines. It is reported that approximately 50% of populations in high income nations have received the vaccines, whereas in low-income nations, barely more than 1% have received any vaccine [12]. This is why we need to be concerned with vaccine inequity matters, and the steps that need to be taken to make equal access possible.

From UNDP data [13], there is a clear split among income groups, with lower income groups lagging behind in vaccine delivery [12]. This inequality becomes manifest when issues on a continent-by-continent basis are considered: Asia, Europe and North America are ahead of Africa, Oceania and South America [12].

If authorities fail to ensure vaccination in poorer, less vaccinated areas, the world may see the emergence of vaccine-resistant variants, which could threaten the whole global population; which many translate as “nobody is safe until everybody is safe” [12].

Caution has been given to the fact that socio-economic recovery in low- and middle-income nations would be hit by the unfair rollout of vaccines. It is reported that $38 billion could have been added to low-income nations’ GDP projections for the year 2021 if such countries received similar vaccination rates as their high-income counterparts [12].

With regard to the difficulties that confront potential countries which donate the COVID-19 vaccines, there are indications that countries of the developing world are not being assisted in their bids to assess the vaccines early enough [14,15]. Most developed countries have given priority to meeting their health requirements. It has been reported in certain circles that most of these nations had come to agreements to procure vaccines for their citizens [14,16]. Several of these deals are tied with pharmaceutical companies [16]. Such countries would no doubt give precedence to their needs. This increase in expenses made by donor countries has been reported in recent times. Many countries have struck deals involving huge financial commitments [14] by way of responding to COVID-19 and the taking receipt of vaccines when ready. This increases the likelihood of developing countries being left out of the list of priorities by the donor.

Key drivers of economic growth such as education and health, which may affect vaccine acceptance, were not considered in this analysis. Additionally, vaccine import and export control cooperation and conflicts between countries that may have an impact on COVID-19 vaccination acceptance were not considered due to the limitations of the dataset used in this study. However, this current study aimed to establish the association between nations’ GDP with COVID-19 vaccination rates. It has been revealed with COVID-19 vaccination progress that humanity is divided into “haves” and “have-nots”. Developed countries are securing their populations with vaccination, unlike the developing countries. The important takeaway message is that the COVID-19 vaccine rollout is uneven and that countries in the Global South are not performing as well as those in the Global North. Therefore, there are large country-level disparities in being able to achieve herd immunity, and efforts toward attaining global herd immunity are severely hampered. One of the biggest challenges that may be observed in the equitable global distribution of COVID-19 vaccines is that there is a disconnect between how some countries agree to do things to show global unity among leaders and the real implementation of the agreement.

A perfect example is the COVAX facility. COVAX is a global initiative established to ensure the speedy and equitable acquisition of COVID-19 vaccines for all nations, irrespective of income level. Low-income countries obtain support from wealthier ones and other donors for vaccine purchases through the Advance Market Commitment (AMC) to make sure that these countries can access the new vaccines at the same time as the lower-middle, middle, upper-middle, and high-income countries. However, in practice, the scenario is a bit different, which was the fear of the World Health Organization (WHO) and public health advocates. Internationally, high-income countries have adopted a competitive “fend for yourself” attitude, competing against others for access to supplies and commercial advantage in the COVID-19 vaccines.

Most of the leading vaccines’ supply was pre-ordered by wealthy nations, even before the safety and efficacy data was made accessible. Therefore, the nationalistic competition for vaccines is a key factor contributing to the challenges faced in the equitable global distribution of COVID-19 vaccines. These practices are contrary to the global interest and are likely to harm countries and citizens of the Global South. Even in Ghana’s situation, where the country (which became the first to receive a shipment of the vaccine from the COVAX initiative, but, currently, the supply is just enough for one percent of the country’s population) recently received 600,000 doses of the AstraZeneca/Oxford vaccine, it is seen that this is still too low of a supply to be able to achieve herd immunity. The global pattern observed is that the low-income countries with the least economic and political bargaining power have less access to vaccines.

There is an urgency to adopt an attitude, an approach, and actions that reflect global fairness, solidarity, and equity for the expedient vaccine distribution and immunisation campaigns across the globe in order that there is not a discrepancy between words and actions. The WHO director-general, Dr. Tedros Adhanom Ghebreyesus, stated it clearly in his opening speech of a WHO executive board meeting: “Not only does this me-first approach leave the world’s poorest and most vulnerable people at risk, it is also self-defeating. Ultimately, these actions will only prolong the pandemic, prolong our pain, the restrictions needed to contain it, and human and economic suffering” [17]. This recommendation also resonates with the idea that no one is safe until everyone is safe. There is a risk of exacerbating more inequities in COVID-19 infection and mortality rates with some of the current practices.

Furthermore, there is the need for local, national, regional, and international coordination of the vaccination rollout. There is already an established mechanism to actualise this. It is known as the Access to COVID-19 Tools (ACT) Accelerator, a partnership launched by WHO and its partners to support this coordinated and global effort [18].

## 5. Conclusions

There is a need for both a top-down and bottom-up approach; that is, a strong commitment, cooperation, and implementation of plans among our scientific, industrial, and political leaders in conjunction with community mobilization at the local levels. The important message driving these recommendations is that, given that the COVID-19 pandemic has effects on a global scale, there is a need for a global response instead of a one-region at a time high-income-countries-before-all others type of response to control the COVID-19 pandemic. The wealthier nations will not be secured without adequately vaccinating the poor ones. Therefore, the global community should take initiatives to speed up the COVID-19 vaccination program in all countries of the world, irrespective of their wealth.

## Figures and Tables

**Figure 1 vaccines-10-00266-f001:**
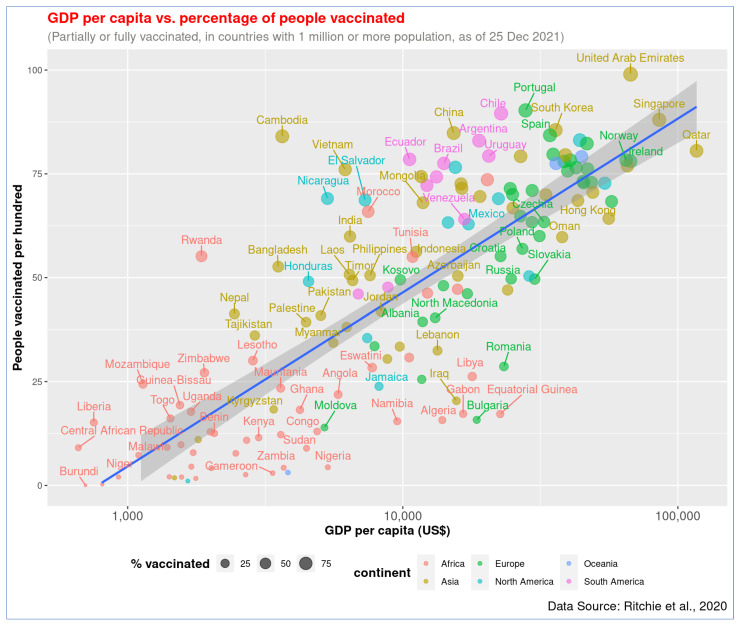
Association of GDP per capita and COVID-19 vaccination rate (full or partial) across countries. The regression line has been presented with blue color. The figure shows that wealthier nations have higher vaccination rates.

**Figure 2 vaccines-10-00266-f002:**
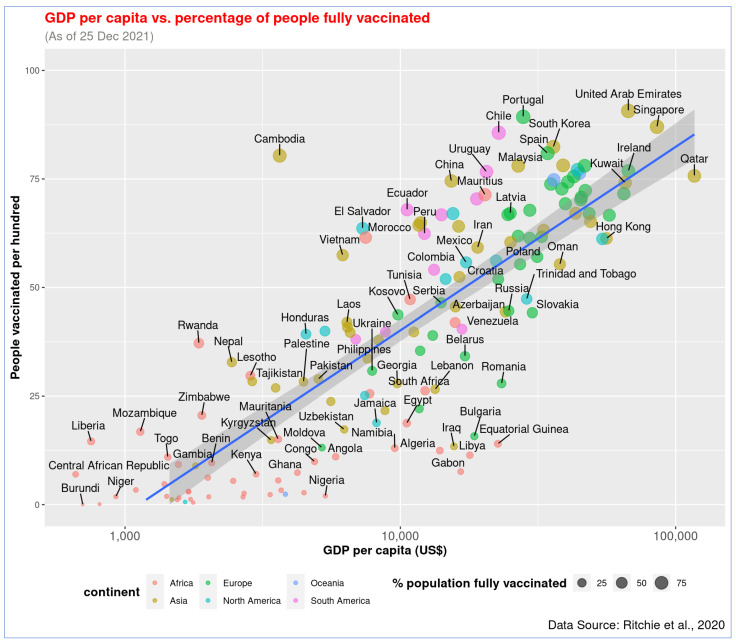
Association of GDP per capita and COVID-19 full vaccination rate across countries. The relationship of both partial and full vaccination rates with GDP is similar.

**Figure 3 vaccines-10-00266-f003:**
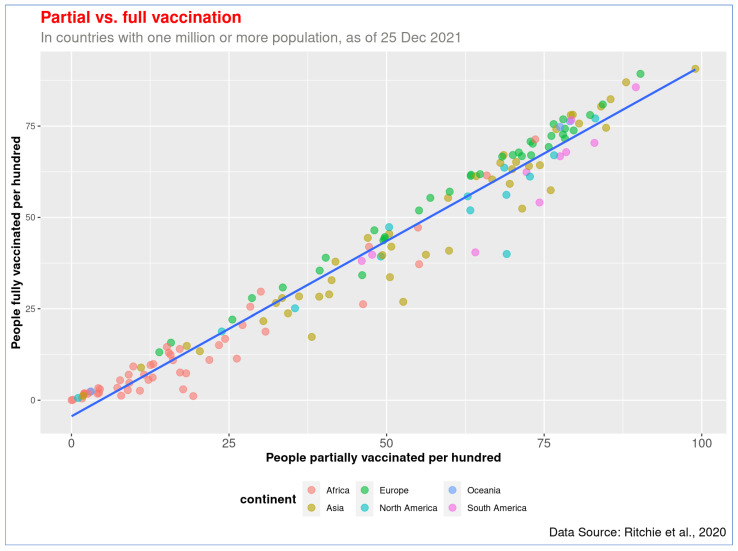
Association between partial and full vaccination. Countries that have achieved a higher level of partial vaccination also reached a higher full vaccination, and most African countries are on the lower left side of the graph.

**Figure 4 vaccines-10-00266-f004:**
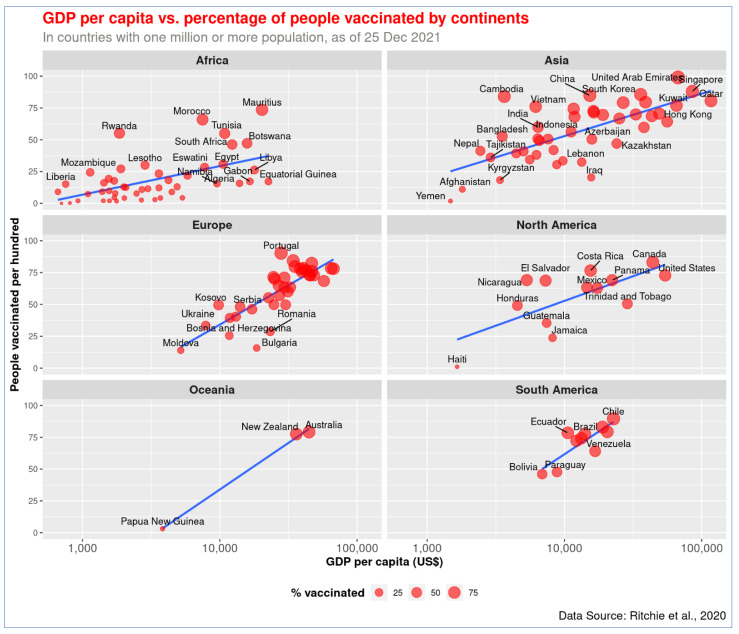
Association of GDP per capita and COVID-19 vaccination rate (full or partial) by continents. The figure shows that wealthier nations got higher vaccination rates on all continents.

**Figure 5 vaccines-10-00266-f005:**
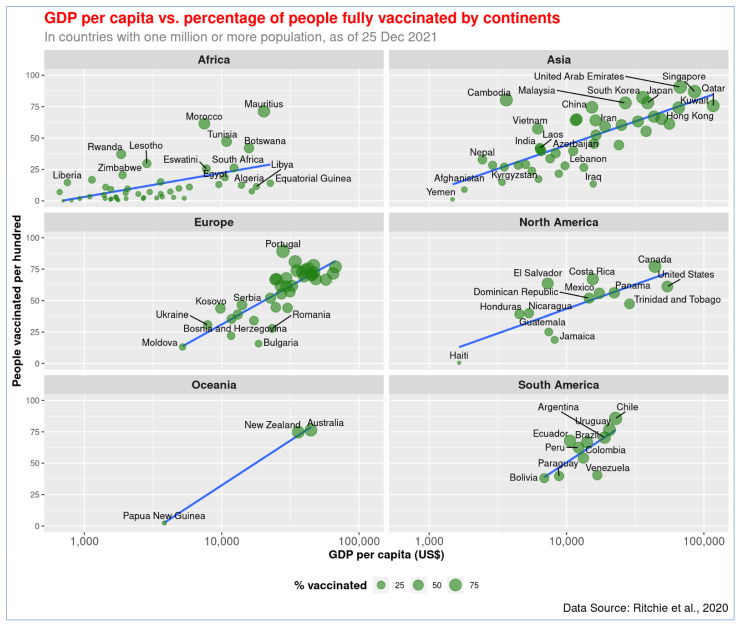
Association of GDP per capita and COVID-19 full vaccination rate across continents. The relationship of both partial and full vaccination rates with GDP is similar in the continents.

**Figure 6 vaccines-10-00266-f006:**
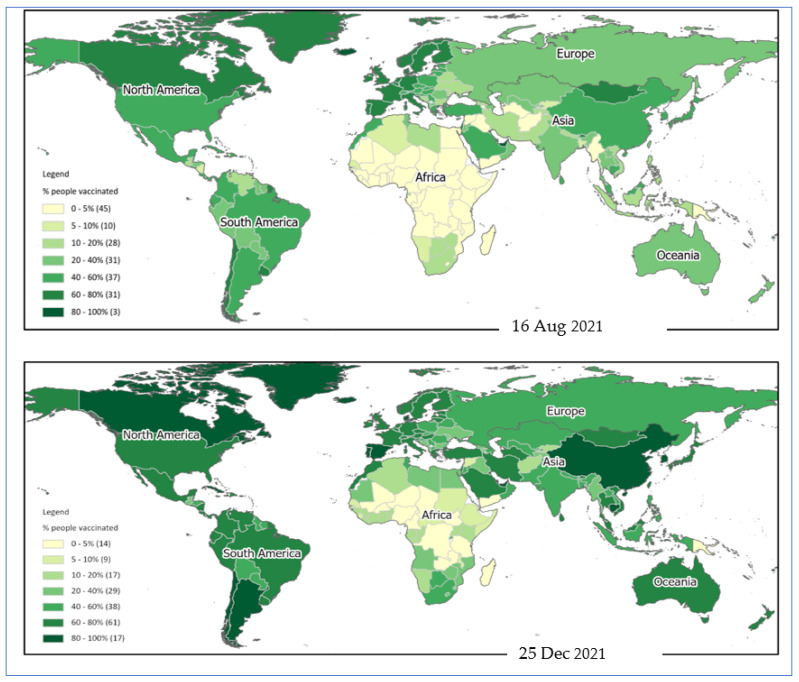
Spatial distribution pattern of COVID-19 vaccination during August and December 2021. Countries in the American and European continent made the highest progress in providing at least one dose of vaccination to their population. Low-income African countries are still lagging behind.

## Data Availability

The dataset of this study can be downloaded at: https://rstudio.cloud/project/2771953 (accessed on 25 December 2021).

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
