# Peer review of "A Global Study on the Correlates of Gross Domestic Product (GDP) and COVID-19 Vaccine Distribution"

_vaccines, 2022, doi:10.3390/vaccines10020266_

Round 1

Reviewer 1 Report

The paper is well written and publishable. The data collected well support the conclusion. The topic is timely and cutting-edge, and once published, will attract wide readship.

However, there may be some room for improvement. First, as between Global North and South, the paper simply focuses on Africa without referring to other regions such as Asia, or Asia-Pacific. It is most likely that there must be differences between Africa and Asia so that there is a need of further research. Second, as for nationalism, it is not very much touched upon in the paper. The paper fouses mainly on GDP, rather than nationalism which needs to be further elaborated.

Anyway, it is a very decent paper. With some necessary revisions, its quality should be further heightened.

Author Response

The paper is well written and publishable. The data collected well support the conclusion. The topic is timely and cutting-edge, and once published, will attract wide readship.

However, there may be some room for improvement. First, as between Global North and South, the paper simply focuses on Africa without referring to other regions such as Asia, or Asia-Pacific. It is most likely that there must be differences between Africa and Asia so that there is a need of further research.

Response: Agreed and we have adjusted the discussion section to reflect your suggestions.

Second, as for nationalism, it is not very much touched upon in the paper. The paper fouses mainly on GDP, rather than nationalism which needs to be further elaborated.

Anyway, it is a very decent paper. With some necessary revisions, its quality should be further heightened.

Response: Agreed and the title has changed to

“A Global Study on the Correlates of Gross Domestic Product (GDP) and COVID-19 Vaccine Distribution”

Reviewer 2 Report

The paper by Palash Basak et al. entitled “Global Perspective of COVID-19 Vaccine Nationalism” is a global study about a correlation between wealth of country and COVID-19 vaccine distribution. The authors found that the vaccination status for COVID-19 was highly skewed across countries, and it correlated with GDP per capita. Although the results of the paper are relatively predictable, there is a certain value in the fact that it presents them as clear data. These findings will be of interest to readers of the journal, however, I have the following concern on the current form.

Comment

  1. The figures are annotated with the names of many countries, but still not all of them are annotated. The data for all countries should be added as a Table or Supplementary Table.

Author Response

The paper by Palash Basak et al. entitled “Global Perspective of COVID-19 Vaccine Nationalism” is a global study about a correlation between wealth of country and COVID-19 vaccine distribution. The authors found that the vaccination status for COVID-19 was highly skewed across countries, and it correlated with GDP per capita. Although the results of the paper are relatively predictable, there is a certain value in the fact that it presents them as clear data. These findings will be of interest to readers of the journal, however, I have the following concern on the current form.

Comment

Response: Agreed and the title has changed to

A Global Study on the Correlates of Gross Domestic Product (GDP) and COVID-19 Vaccine Distribution

  1. The figures are annotated with the names of many countries, but still not all of them are annotated. The data for all countries should be added as a Table or Supplementary Table.

Response: The graphs mainly demonstrate the association between the two variables under consideration. This size of graphs is not suitable for showing all country names. However, as suggested by the reviewer, we have added the dataset for all countries as a supplementary table.

COVID-19 Vaccination Rate, GDP, and Population Data by Countries – as of 25 Dec 2021 07:43pm Sydney time – summarised from Ritchie et al 2020

Reviewer 3 Report

The title is misleading. The paper does not have anything to do with Nationalism which is a political concept. It just looks at the bivariate relationship among GDP and vaccination.

The analysis does not go much beyond what can be found in the ourworldindata page, https://ourworldindata.org/grapher/covid-vaccinations-vs-gdp-per-capita. The only “contribution” is the separate analysis by continent. I don’t think this is enough to make it a separate research article. The analysis is much less detailed because it only looks at a cross-sectional image, not making use of the dynamic information.

The references do not Include and do not benefit from other studies looking at the association between GDP and vaccination. For instance Ngo, Zimmermann et al. (2021) find that the impact of GDP is different at different times in the process of vaccination, becoming more important as the vaccination process progresses. They also control for alternative possible explanations such as education and covid incidence, which have a separate contribution. This is important to separate two possible reasons for lack of progress in vaccination (economic vs acceptance of vaccination).

Of course, the overall message is important but the contribution of this particular manuscript seems minor.

Reference:

Ngo, Vu M.; Zimmermann, Klaus F.; Nguyen, Phuc V.; Huynh, Toan L.D.; Nguyen, Huan H. (2021) : How education and GDP drive the COVID-19 vaccination campaign, GLO Discussion Paper, No. 986, Global Labor Organization (GLO), Essen. http://hdl.handle.net/10419/246812

Author Response

The title is misleading. The paper does not have anything to do with Nationalism which is a political concept. It just looks at the bivariate relationship among GDP and vaccination.

Response: Agreed and title now reads:

“A Global Study on the Correlates of Gross Domestic Product (GDP) and COVID-19 Vaccine Distribution”

The analysis does not go much beyond what can be found in the ourworldindata page, https://ourworldindata.org/grapher/covid-vaccinations-vs-gdp-per-capita. The only “contribution” is the separate analysis by continent. I don’t think this is enough to make it a separate research article. The analysis is much less detailed because it only looks at a cross-sectional image, not making use of the dynamic information.

The references do not Include and do not benefit from other studies looking at the association between GDP and vaccination. For instance Ngo, Zimmermann et al. (2021) find that the impact of GDP is different at different times in the process of vaccination, becoming more important as the vaccination process progresses. They also control for alternative possible explanations such as education and covid incidence, which have a separate contribution. This is important to separate two possible reasons for lack of progress in vaccination (economic vs acceptance of vaccination). Of course, the overall message is important but the contribution of this particular manuscript seems minor.

Reference:

Ngo, Vu M.; Zimmermann, Klaus F.; Nguyen, Phuc V.; Huynh, Toan L.D.; Nguyen, Huan H. (2021) : How education and GDP drive the COVID-19 vaccination campaign, GLO Discussion Paper, No. 986, Global Labor Organization (GLO), Essen. http://hdl.handle.net/10419/246812

Response: We agreed with the reviewer. However, data on the acceptance of vaccination for all countries are not readily available as of now.  Additionally, we have added these as a  key limitation to the study.

Reviewer 4 Report

The purpose of this study is to investigate the association between national GDP and COVID-19 vaccination progress from a global perspective, and to study the spatial distribution pattern of COVID-19 vaccination progress in each country to further analyze COVID-19 vaccine nationalism. However, the theme and content fit of this study are insufficient, and the analysis method is too one-dimensional; it is recommended to enrich the existing analysis framework.

  1. The main thrust of this paper is to dissect COVID-19 vaccine nationalism from a global perspective, but the concept of COVID-19 vaccine nationalism is not defined throughout the paper.
  2. Considering that only the correlation between country GDP and vaccination rates was analyzed, are the results sufficient to support what is discussed in this paper on COVID-19 vaccine nationalism?
  3. The prevalence of vaccination rates is related to many factors; technical investment in national vaccine R&D, vaccine production efficiency, vaccine import/export control, cooperation and conflicts between countries, etc. Is it sufficient to analyze all countries using GDP, without considering factors such as vaccine import/export situation?
  4. The article has a lot of errors in details, which I hope the author will pay attention to. For example, in the first paragraph of the results section, GDP is misspelled as GPD.

Author Response

The purpose of this study is to investigate the association between national GDP and COVID-19 vaccination progress from a global perspective, and to study the spatial distribution pattern of COVID-19 vaccination progress in each country to further analyze COVID-19 vaccine nationalism. However, the theme and content fit of this study are insufficient, and the analysis method is too one-dimensional; it is recommended to enrich the existing analysis framework.

Response:  We agreed with the reviewer but unfortunately, data on other dimensions, such as vaccine acceptance and vaccine import/export, are not readily available. we have added this point as a limitation to our study. Future studies can shed light on this.

  1. The main thrust of this paper is to dissect COVID-19 vaccine nationalism from a global perspective, but the concept of COVID-19 vaccine nationalism is not defined throughout the paper.
  2. Considering that only the correlation between country GDP and vaccination rates was analyzed, are the results sufficient to support what is discussed in this paper on COVID-19 vaccine nationalism?
  3. The prevalence of vaccination rates is related to many factors; technical investment in national vaccine R&D, vaccine production efficiency, vaccine import/export control, cooperation and conflicts between countries, etc. Is it sufficient to analyze all countries using GDP, without considering factors such as vaccine import/export situation?

Response: We agreed with the reviewer but Global-level data for vaccine import/export is unavailable as of now. Hence, we have added this point as a limitation of our study. Future studies can shed light on that.

  1. The article has a lot of errors in details, which I hope the author will pay attention to. For example, in the first paragraph of the results section, GDP is misspelled as GPD.

Response: We have revised the manuscripts for the minor errors.

Round 2

Reviewer 3 Report

If the previous version was, in my opinion, not up to the standard of publication of vaccines, this one is even less so.

The authors admit the limitations of the analysis but have lengthened the discussion section making the kind of claims that they acknowledge the analysis does not allow to do. You cannot make causal analysis from a cross-section. You have not tried to identify to what extent the lack of vaccination is demand or supply driven.

The discussion would be more suited in a letter to the editor format. It is not really a scientific paper making a contribution.

The supplementary appendix is unprofessional. If you only distribute the data, do not do it in a pdf format, which is not suitable. You can always redistribute the csv exactly as it came from ourworldindata.org. At least that is useable. Of course not original, but there is nothing original in this analysis, except for interpretations that do not stem from the analysis.

Author Response

Query 1:The authors admit the limitations of the analysis but have lengthened the discussion section making the kind of claims that they acknowledge the analysis does not allow to do. You cannot make causal analysis from a cross-section. You have not tried to identify to what extent the lack of vaccination is demand or supply driven.

Response: We appreciate the reference you suggested in your first review but we want to avoid “comparing oranges with apples” by not merging the two data sets because the data set used in this study was collected at certain time (as of 25 Dec 2021 07:43pm Sydney time) whereas from the https://www.econstor.eu/esstatistics/10419/246812?year=2021&month=12 website, it is NOT possible for us to get the same data set at that same time. Additionally, information on key variables such as years of schooling index (UNDP, 2020) and educational system (YearsofSchooling) were proximate estimates which could be another limitation of using this data set.

From Epidemiological viewpoint, you cannot make any causal analysis from two data bases collected at different time, this could lead to huge bias we cannot account for statistically such time differences, proxy estimates of key factors and others.

Query 2: The discussion would be more suited in a letter to the editor format. It is not really a scientific paper making a contribution.

Response: We disagreed with reviewer on this point because the pre-print of this article has been posted on twitter and other social media, we got very encouraging comments, feedback from editors from other high quality journal.

The supplementary appendix is unprofessional. If you only distribute the data, do not do it in a pdf format, which is not suitable. You can always redistribute the csv exactly as it came from ourworldindata.org. At least that is useable. Of course not original, but there is nothing original in this analysis, except for interpretations that do not stem from the analysis.

Response: we have provided PDF file so that we can refer to it as supplementary table in the manuscript. In additional there are a lot of free software on google that can convert PDF to csv, but we have added a csv file as supplementary dataset.

Reviewer 4 Report

I like to thank the authors as they both improved the manuscript after taking all my suggestions into consideration and provided me with a satisfying reply (in the response letter). Hence, I'm pleased to recommend the editor to accept manuscript's current version for publication.

Author Response

Thank you